# The Global Evolutionary History of Orf Virus in Sheep and Goats Revealed by Whole Genomes Data

**DOI:** 10.3390/v16010158

**Published:** 2024-01-21

**Authors:** Elisabetta Coradduzza, Fabio Scarpa, Angela Maria Rocchigiani, Carla Cacciotto, Giada Lostia, Mariangela Stefania Fiori, Yoel Rodriguez Valera, Alessandra Mistral De Pascali, Martina Brandolini, Ilenia Azzena, Chiara Locci, Marco Casu, Roberto Bechere, Davide Pintus, Ciriaco Ligios, Alessandra Scagliarini, Daria Sanna, Giantonella Puggioni

**Affiliations:** 1Istituto Zooprofilattico Sperimentale della Sardegna, 07100 Sassari, Italy; elisabetta.coradduzza@izs-sardegna.it (E.C.); angelamaria.rocchigiani@izs-sardegna.it (A.M.R.); giada.lostia@izs-sardegna.it (G.L.); mariangela.fiori@izs-sardegna.it (M.S.F.); roberto.bechere@izs-sardegna.it (R.B.); davide.pintus@izs-sardegna.it (D.P.); ciriaco.ligios@izs-sardegna.it (C.L.); giantonella.puggioni@izs-sardegna.it (G.P.); 2Dipartimento di Scienze Biomediche, Università di Sassari, 07100 Sassari, Italy; fscarpa@uniss.it (F.S.); iazzena@uniss.it (I.A.); c.locci3@phd.uniss.it (C.L.); 3Dipartimento di Medicina Veterinaria, Università di Sassari, 07100 Sassari, Italy; ccacciotto@uniss.it (C.C.); marcasu@uniss.it (M.C.); 4Mediterranean Center for Disease Control, 07100 Sassari, Italy; 5Faculty of Agricultural Sciences, University of Granma, Bayamo 95100, Cuba; yrodriguezvalera@udg.co.cu; 6Dipartimento di Scienze Mediche e Chirurgiche, Università di Bologna, 40138 Bologna, Italy; alessandra.depascal3@unibo.it (A.M.D.P.); martina.brandolini3@unibo.it (M.B.); alessand.scagliarini@unibo.it (A.S.)

**Keywords:** ORFV, phylodynamics, whole genome sequencing, S genomes, G genomes, molecular dating

## Abstract

Orf virus (ORFV) belongs to the genus Parapoxvirus (Poxviridae family). It is the causative agent of contagious ecthyma (CE) that is an economically detrimental disease affecting small ruminants globally. Contagious ecthyma outbreaks are usually reported in intensive breeding of sheep and goats but they have also been reported in wildlife species. Notably, ORFV can infect humans, leading to a zoonotic disease. This study aims to elucidate the global evolutionary history of ORFV genomes in sheep and goats, including the first genomes from Central America in the analyses. In comparison to the last study on ORFV whole genomes, the database now includes 11 more sheep and goat genomes, representing an increase of 42%. The analysis of such a broader database made it possible to obtain a fine molecular dating of the coalescent time for ORFV S and G genomes, further highlighting the genetic structuring between sheep and goat genomes and corroborating their emergence in the latter half of 20th century.

## 1. Introduction

Orf virus (ORFV) belongs to the genus Parapoxvirus (Poxviridae family), and it is the causative agent of contagious ecthyma [1]. To date, the International Committee of Taxonomy of Viruses (ICTV, 2022) has included four established species of enveloped double-stranded DNA viruses in this genus, which affect domestic and wild small ruminants: ORFV, endemic in most sheep and goat-raising countries; Bovine papular stomatitis virus (BPSV) and Pseudocowpox virus (PCPV), mainly found in cattle with manifestation on muzzle and teats, respectively; and the Red deerpox virus (RDPV), which causes relevant infections in cervids [2]. Additionally, parapoxviruses have been reported to affect camels [3], reindeer, and muskox [4,5]. Similar lesions have also been reported in seals, but the viral DNA appears to be strongly different from that of other parapoxvirus species. Thus, the virus affecting seals (grey sealpox virus) is considered a separate species within the genus Parapoxvirus [6,7]. Contagious ecthyma is spread worldwide, and outbreaks are usually reported among intensive breeding of sheep and goats, causing significant economic losses. However, it has also been reported in other animal species [8] and humans [1,9], thus resulting in a zoonotic disease. During outbreaks, the degree of morbidity is very high, with mortality rate is around 1%; due to secondary infections, the mortality can increase up to 10% in lambs [10] and to 93% in kids [11,12]. In ruminants, lesions usually involve the mouth, muzzle, nostrils, gums, tongue, occasionally the feet and udders, and sporadically the gastrointestinal tract and the respiratory apparatus [13]. In lambs and kids, injuries can cause their death when avoid milk-sucking from their mothers. In humans, ORFV infection usually provokes an occupational disease, mostly affecting workers with direct contact with animals (farmers, veterinarians, butchers, hunters). Lesions are reported mainly on hands and other uncovered areas of the body and undergo spontaneous benign resolution in a few weeks. However, malignant cases with atypical proliferative lesions, affecting mainly immunocompromised individuals, have been also described [8,14,15]. The ORFV genome is a linear, double-stranded, 135 kb-long DNA that encodes 132 genes [8,16]. The central region contains highly conserved genes involved in viral replication and the genesis of the viral structure; conversely, the terminal regions are more variable and contain genes for virulence and immunomodulation [8,17]. Furthermore, the terminal regions are characterized by high nucleotide variability, which could be involved in the reinfection phenomenon typical of ORFV, along with the regulation of virulence genes by the innate immune system of the infected subject [14]. At present, a limited number of studies have described ORFV whole genome variations (i.a., those with phylogenetic purposes, [18,19,20,21,22,23,24]), isolated from Asia, Europe, Oceania, and North America. In general, these studies are consistent with previous research based on the analyses of single genes (i.a., [25,26,27,28,29]) in indicating the occurrence of a genetic structuring of ORFV strains between sheep and goats. These findings suggest that the virus has evolved differently in the two host species, with different evolutionary histories that might have influenced the development of pathogenesis infection in sheep and goats [21,29]. Among these studies, only one [21] has reported a comprehensive inference on phylogeny and phylodynamics of ORFV sheep and goat genomes from all over the world, providing hints on the evolutionary history that characterized their diffusion in Europe and in the Mediterranean island of Sardinia. Based on 26 genomes, it was found that the two ORFV genome types (S and G from sheep and goats, respectively) originated in Europe and Asia in the late 1800s. In particular, the sheep genome that were isolated in Sardinia [21,29] were shown to be exclusive to this Mediterranean island and likely originated by genetic drift from a few European founders.

The global evolutionary dynamics of ORFV remain poorly understood. Therefore, this study aims to perform a phylodynamics study at the highest possible resolution using all sheep and goats ORFV whole genomes available to date. To achieve this goal, the analyses will also include the first genomes from Central America, specifically from Cuba. In the Republic of Cuba, knowledge about the spreading of ORFV is still limited, with reported outbreaks of contagious ecthyma between 2006 and 2009 [30]. The inclusion of genomes from this island in the analyses would be expected to provide additional insights into the worldwide phylogeographic patterns of ORFV diffusion.

## 2. Materials and Methods

### 2.1. Sampling

Pathological samples (see Table 1) were collected between 2007 and 2009 from two naturally infected sheep and one goat during different outbreaks occurring in different municipalities of the eastern provinces of Cuba (Figure 1).

### 2.2. Cell Culture and Viral Infection

Pathological material, represented by scabs collected from diseased sheep (samples 44 and 45) and one goat (sample 33), was inoculated onto one African green monkey kidney epithelial cell line (VERO E6; ATCC CRL-1587). VERO-E6 were cultured in d-MEM (EuroClone, Pero (MI), Italy), supplemented with 10% foetal bovine serum, 1% antibiotic (100 U/mL penicillin, 100 µg/mL streptomycin, EuroClone), and 1% l-glutamine (2 mM, EuroClone), in a 6-wells plate at 37 °C with 5% CO_2_ and 90% humidity. Scabs were grinded using a pestle; resuspended in d-MEM with 10% antibiotic, 2% fetal bovine se-rum, and 1% l-glutamine; and filtered through a 0.45 µm filter before being inoculated onto confluent monolayers. Following a 1 h adsorption step at 37 °C, inoculated cell monolayers were washed with sterile PBS (Phosphate Buffer saline, EuroClone) and incubated with fresh culture medium supplemented with 10% antibiotic, 2% fetal bovine serum, and 1% l-glutamine. Four blind passages have been carried out, lasting approximately 3 days each, until the appearance of a typical cytopathic effect characterized by plaque formation. Subsequently, the isolated viruses from third-passage supernatants were titrated using the 50% tissue culture infectious dose (TCID50) endpoint dilution assay and genomic material was identified using PCR targeting B2L gene [31] upon viral DNA extraction with QIAmp UltraSens Virus Kit (Qiagen, Hilden, Germany).

### 2.3. Viral DNA Extraction, Sequencing, and Genome Assembly

In order to confirm the presence of ORFV, the DNeasy Blood and Tissue Kit (Qiagen) was used to extract DNA from 25 mg of each lesional tissue sample, as described in the user’s manual. The VIR gene was then amplified using the VIR1 and VIR2 primers [32] that provided fragments of 617 bp. Moreover, a further set of primers (VIR 3 and VIR 4) was designed to amplify a fragment of approximately 817 bp in samples VIR negative to the first PCR. PCR was performed according to the protocol described by Kottaridi et al. [33], with slight changes in annealing temperatures [21]. To test the effectiveness of PCR protocols and the absence of possible contamination, positive (old high-quality ORFV DNA samples that always give good results when amplified) and negative controls were used. Electrophoreses was carried out using the Invitrogen E-Gel EX 2% agarose kit (gel precast). Specific DNA PCR bands were excised from the gel and purified using the QIAquick Gel Extraction Kit (Qiagen). In the presence of non-specific bands during electrophoresis, the CleanSweep PCR Purification Kit (Thermo Fisher Scientific, Waltham, Massachusetts, Stati Uniti) was used to purify the PCR products. The resulting PCR products were then sequenced for both forward and reverse strands (using the same primers used for PCR) using a Sanger Sequencing 3500 Series Genetic Analyzers Terminator 3.1 apparatus (Applied Biosystems, Waltham, Massachusetts, Stati Uniti). The whole genome sequencing was performed using the Illumina platform as described by Fiori et al. [34] on viral DNA extracted from VERO cell culture supernatant by QIAmp Ul-traSens Virus Kit (Qiagen, Hilden, Germany). DNA quantification was performed using an Epoch microplate spectrophotometer (BioTek, Winooski, VT, USA) and a Qubit 2.0 Fluorometer (Thermo Fisher Scientific, Waltham, MA, USA), agreeing with the manufacturer’s instructions. Viral DNA libraries were prepared using the Nextera DNA Flex Library Prep kit (Illumina Inc., San Diego, CA, USA). The samples’ whole-genome shotgun (WGS) sequencing was performed using the Illumina MiSeq platform generating paired-end reads 2 × 300 (external core service BMR Genomics, Padua, Italy). Samples have been sequenced through the WGS technique on the MiSeq platform by BMR Genomics in Padua, Italy, generating paired end reads 2 × 300 (Illumina, San Diego, California, Stati Uniti). The resulting FASTQ files were checked for quality using the tool fastQC from the FASTX-Toolkit v0.7 [35]; assembly with reference and de novo were performed with Bowtie 2 v2.4.2 (Johns Hopkins University, Baltimore, MD, USA) [36,37] and SPAdes v3.15.3 (Center for Algorithmic Biotechnology, St. Petersburg, Russia) [38] following Scarpa et al. [39] and Piredda et al. [40]. Scaffolds were aligned with the reference genome NC_005336.1 using Unipro UGENE v.35 (Unipro Center for Information Technologies, Novosibirsk, Russia) [41]. The newly sequenced genomes have been deposited in GenBank (refer to Table 2 for accession numbers).

### 2.4. Phylodynamics and Molecular Dating

Three ORFV whole genome sequences were obtained for ORFV in the present study from isolates collected in the South of the island of Cuba. Two strains were isolated from sheep in the Province of Guantanamo and Granma, and one strain was isolated from a goat in the province of Guantanamo.

These new three genome sequences were included in a large dataset containing all the ORFV whole genomes available on GenBank to date, considering only sheep and goat-derived strains (see Table 2 for details and references). Noteworthily, the strains isolated in humans were excluded from the analyses to deeply infer the evolutionary patterns of ORFV genomes isolated from sheep and goats separately. Therefore, the dataset that was obtained included a total of 37 sequences: 7 from Italy, 2 from Spain, 13 from China, 3 from India, 2 from Malaysia, 1 from New Zealand, 4 from the United States of America (USA), 2 from Argentina, and 3 from Cuba (from the present study).

The complete dataset underwent alignment using the L-INS-I algorithm within Mafft 7.471 [44] and was subsequently manually inspected and refined utilizing Unipro UGENE v.35 [41]. Principal Coordinates Analysis (PCoA) was conducted through GenAlEX 6.5 [45], utilizing a pairwise p-distance matrix of genetic data. This analysis aimed to identify potential subgroups within genetic clusters and evaluate the dissimilarities reflecting genetic variations among sequences [46]. The jModeltest 2.1.1 [47] software was utilized to determine the most suitable probabilistic model for genome evolution, employing a maximum likelihood optimized search. For Bayesian phylogenetic analysis, MrBayes 3.2.7 [48] was employed, with specific model parameter settings (nst = 6, rates = invgamma, ngammacat = 4). Two independent runs, utilizing 4 Metropolis Coupled Markov Chain Monte Carlo (MCMCMC) chains (1 cold and 3 heated chains), were conducted over 5 million generations, discarding the initial 25% of sampled trees for burn-in. Convergence of chains and model reliability were assessed based on the average standard deviation of split frequencies [48] and the potential scale reduction factor [49], following Scarpa et al. [50].

Phylogenetic trees were visualized and edited using FigTree 1.4.1 (available at http://tree.bio.ed.ac.uk/software/figtree/). Molecular dating, when feasible based on sample collection dates (at least month and year), was performed using a Bayesian approach under the MCMC algorithm within Beast 1.10.4 software [51]. Only genomes with available collection dates were included for this analysis. Different clock models (strict and uncorrelated log-normal relaxed) were tested via rapid runs of 100 million generations, selecting the optimal model based on Bayes factor values from Tracer 1.7 software [52]. Various demographic models were also evaluated.

Phylogenetic time-scaled trees and evolutionary rates were co-estimated, following the selection of the Bayesian skyline demographic model under the uncorrelated log-normal relaxed clock model, running 500 million generations with sub-sampling every 50,000 generations. Resulting log files were scrutinized with Tracer 1.7 [52], accepting only values of ESS (effective sample size) ≥ 200. The maximum clade credibility tree was generated using TreeAnnotator (Beast package) and visualized using FigTree 1.4.1.

Furthermore, Beast software was utilized for additional runs under the coalescent Bayesian skyline demographic model [53] to assess variations in population size over time relative to genetic variability, trace lineage expansions, and estimate evolutionary rates for ORFV strains isolated in sheep and goats.

## 3. Results

### 3.1. Virus Isolation

ORFV was isolated in VERO E6 cells from all propagated specimens. Cytopathic effect was detected on the confluent monolayer approximately 3 days after infection including rounding and ballooning, degeneration and detachment (Figure 2). Active replication was observed on the basis of the decrease in ct from the first and third passages.

### 3.2. Molecular Virus Identification

Cuban samples after testing positive to ORFV VIR gene, were successfully sequenced, and the first three whole genomes for Central America were successfully obtained. Assembly was performed for these three ORFV strains with the reference genome (NC_005336).

### 3.3. Phylodynamics and Molecular Dating

In the present study, a 131,449 bp-long sequences alignment, including a total of 38 ORFV whole genomes from sheep and goats, was investigated. These sequences were isolated in all the continents with the exception of Africa (see Table 2 for details). In the dataset, three genomes from Cuba, which were isolated in the present study, represent the first ORFV strains from Central America. Indeed, for this large area of the Americas, no molecular data are known for this virus so far. The nucleotide frequency analysis carried out on the whole dataset revealed that conserved sites amount to 71,418 (54.3%), variable sites amount to 65,674 (50%), parsimony informative sites amount to 15,396 (11.7%), and singletons amount to 49,812 (37.9%).

The PCoA (Figure 3 and Table 3) was performed to confirm the occurrence of a genetic structuring between ORFV genomes from sheep and goats as it was reported by previous studies (i.a., [18,19,20,21,22,29]). The first two axes of the analysis were able to explain 68.21% of the variability (PCoA1/X axis: 35.93%, PCoA2/Y axis: 32.28%). Results pointed out the presence of a divergence along the first axis between ORFV strains isolated from sheep and goats (cluster S and cluster G, respectively, in Figure 3). The only relevant exception to this pattern was represented by two genomes isolated from sheep, which sets within the cluster of goats. These genomes were from Cuba (Guantanamo province) and China, and the possibility that they may correspond to divergent goat viral strains likely able to infect severely immunocompromised sheep, rather than anomalies to the genetic structuring between S and G, cannot be ruled out.

Furthermore, it is interesting to note that in the PCoA graphic, one genome isolated in a goat from China is included as an outlier outside the variability of the strains isolated from sheep.

The phylogenetic mid-point tree analysis evidenced the occurrence of two fully supported genetic clades (clade S and clade G in Figure 4). Results obtained from the Bayesian phylogenetic time-scaled maximum clade credibility tree (ultrametric tree), which are not displayed here, are generally consistent in the main topology of the tree with those obtained from the standard phylogenetic tree. Any minor discrepancies observed can be attributed to the use of different algorithm versions, with the software Beast employing MCMC and MrBayes employing the metropolis coupled implementation (MCMCMC). The coalescence time at each node of the phylogenetic tree was inferred according to the molecular dating estimates obtained with the ultrametric tree with the confidence interval (C.I.) at 95% of the highest posterior density (HPD). Only genomes whose sample collection date was available were used to perform the molecular dating.

The clade G of the phylogenetic tree (Figure 4), which dates to the year 1947, includes only ORFV genomes isolated from goats, hereinafter named as G genomes. The only exceptions are represented by one genome isolated from a sheep in Cuba and two genomes isolated from sheep in China.

In the clade G of the phylogenetic tree, four main internal groups, which include sequences according to their geographical origin, are present. These groups form two main sister clusters within clade G. The first sister cluster is representative of North America and Asia. The group of sequences from North America originated in 1982 and includes two genomes from USA. The group of sequences from Asia originated in 1960 and groups genomes from China, India, and Malaysia. The second sister cluster is representative of South Europe and Central America. The group of South Europe dates to 2018 and includes the only European genomes of clade G that correspond to two strains isolated in the Mediterranean Sardinia Island. The group of Central America originated in 2001 and includes two strains isolated in the Guantanamo province of the island of Cuba. Noteworthily, one of these latter Cuban genomes, was isolated from a sheep and corresponds to one of the two ORFV strains from sheep that laid within the G-genomes cluster of the PCoA (see Figure 3 and Table 3). Noteworthily, within clade G, a further external cluster is present. It dates back to 1990 and contains three Chinese genomes, one of them corresponds to the sequences, which was set as an outlier in the PCoA (see Figure 3 and Table 3), while the remaining genomes were isolated from sheep.

The clade S of the phylogenetic tree, which originated in 1957, includes only sequences isolated in sheep, hereinafter named as S genomes, with only one exception being represented by a Chinese genome. This clade includes two internal sister clusters, the largest of them (indicated as Global in Figure 3) originated in 1966 and includes sequences from Asia (China), Oceania (New Zealand), North (USA), Central (from the Granma province of Cuba), and South America (Argentina), as well as South Europe (Spain). Within this cluster, a sequence isolated in China in a goat is also present; interestingly, this sequence is part of a little cluster that dates back to 2017 and includes another Chinese sequence, isolated in a sheep, that laid within the G-genome cluster in the PCoA (see Figure 3 and Table 3). The second sister group of clade S, which originated in 1963, includes only South European sequences that were isolated in the Sardinia Island.

Interestingly, one sequence isolated in Sardinia in 2021 sets in a basal position, external to the whole clade S.

The ORFV G genomes’ patterns of spreading among goats were inferred by the Bayesian skyline plot (BSP in Figure 5a) that showed a long-lasting constant size diffusion from their origin with a weak exponential increase during the first 25 years after the differentiation. A drastic decrease in the ORFV G genomes’ spread was found after 2019. Consistently, the analysis of the lineages through time (LTT in Figure 5b), also evidenced that, from the first radiation of G genomes among hosts, the amount of the lineages experienced a constant size expansion, with a further slight increase after 2019.

A consistent trend was obtained for ORFV S genomes. Indeed, their patterns of spreading among sheep, inferred by the Bayesian skyline plot (BSP in Figure 6a), showed a long-lasting constant size diffusion from their origin with a weak exponential increase during the first 10 years after the differentiation. A drastic decrease in the ORFV S genomes’ spread was found after 2006, which lasted for about 5 years, being quickly followed by short new exponential growth. The analysis of the lineages through time (LTT in Figure 6b) also evidenced that, from the first radiation of S genomes among hosts, the amount of their lineages experienced a constant exponential expansion, whose extent steadily decreased until it reached a plateau in 2016.

## 4. Discussion

Contagious ecthyma, caused by Orf virus, has a worldwide diffusion, and despite the zoonotic potential and the economic impact, it still represents an almost neglected zoonotic disease [54]. Considering the contribution of the present study, to date, about 40 ORFV complete genomes are available in public databases. In this framework, the data reported here represent a further step forward in filling knowledge gaps on ORFV, as it provided the first genomes from Central America, thus expanding knowledge, with three strains isolated from two sheep and one goat in the Republic of Cuba, the number of countries for which ORFV genomes are known.

We used a molecular approach which combines the tools of phylodynamics analyses to techniques of phylogeny and phylogeography to depict the global evolutionary history and perform molecular dating for the first time for the two main ORFV lineages (S and G genomes).

Results derived from our analyses corroborate the occurrence of the genetic structuring between genomes of ORFV isolated from sheep and goat as it was suggested by previous studies (i.a., [18,19,20,21,22,29]). The G-genomes’ clade includes different sub-clusters that are likely exclusively of distinct large geographic areas (i.e., North America, Asia, Europe, and Central America). Conversely, a weak geographic structuring is present for the S-genomes’ clade, with the only exception of two groups of sequences from China and the Mediterranean Sardinia Island (Italy).

It is important to note that the absence of ORFV genomes from northern Europe in the analyzed dataset may introduce a bias in the results, emphasizing the need for future analyses including data from this region to validate or refine the evolutionary trend observed in the present study.

The analysis of more than 42% additional sheep and goat genomes (11 new genome sequences) compared to Coradduzza et al. [21], together with the exclusion of the genomes isolated in humans from the dataset, allowed us to obtain a more precise molecular dating of the coalescent time for the ORFV S and G genomes.

Our findings are consistent with Coradduzza et al. [21], indicating that these modern genomes originated around the 1950s. These years coincided with the post-Second World War period, marked by increasing human and animal movements and demographic expansion. The overlapping of habitats for humankind and animals led to the global influenza pandemic [55], producing a scenario resembling the spread ORFV.

In our analyses, a discrepancy with the coalescent time estimates for S and G genomes provided by Coradduzza et al. [21] was recorded, probably due to the presence in the previous study of ORFV genomes isolated in humans that likely represent evolutionary dead ends without descendants. Indeed, the presence in the analysis by Coradduzza et al. [21] of the human strains allowed us to calculate the divergence time for the common ancestor to both modern S and G genomes and strains that could infect animals and humans as well. These latter strains could have differentiated before S and G genomes from a common ancestor. Nonetheless, they did not have an evolutionary pathway which produced many descendants as it occurred for S and G genomes, as a possible consequence of a reduced fitness, which prevented its adaptation and spread among a large number of hosts.

Although S and G genomes are not exactly coeval (the genomes exclusive to sheep differentiated in about 1953, whereas those exclusive to goats differentiated in about 1947), our results suggested, for the first time, the same evolutive model for these two types of ORFV genomes.

Noteworthily, our results pointed out a decrease in the spread of viral strains for S genomes corresponding to an increase in the amount of viral lineages starting from the year 2006. This specific evolutionary trend is representative of a rapid viral adaptation process, generally due to the decrease in the number of hosts that produces the need for the virus to evolve quickly by increasing its infection capability. Interestingly, in 2006–2007, some viral epidemics occurred in Europe, Africa and Asia, producing the death of hundreds of thousands of sheep (i.a., [56,57,58,59,60,61,62,63,64]). Although perhaps biased by the low number of available sequences for that period, this sheep worldwide mass mortality could have provoked the consequent decrease of ORFV S-genomes thus boosting this virus to quickly evolve producing new lineages to survive.

As well, our results evidenced a decrease in the spread of for G genomes corresponding to an increase in the amount of viral lineages starting from the year 2019. Interestingly, the World Organization for Animal Health reported, for the years 2019–2020, the occurrence of a relevant goat mass mortality was likely caused by a combination of factors (e.g., diseases, starving) (i.a., [63,65]). In the light of the scant number of available sequences for the period, such a goat mortality event might have provoked the ORFV G genomes’ decrease among hosts and consequently prompted the exponential increase of new lineages to cope with the death of a huge number of hosts.

Our results also highlighted the occurrence of viral strains that have been isolated in hosts different from those specific for their genomic lineage. Indeed, PCoA and the phylogenetic tree (Figure 3 and Figure 4) evidenced ORFV genomes isolated from sheep in China and Cuba within the cluster of G genomes, and, vice versa, Chinese genomes isolated from goats within the cluster of S genomes. Remarkably, one of these latter Chinese genomes isolated in a goat (GB#: MN648218) was already found included within the S genomes clade by Coradduzza et al. [21]. Paying due attention in interpreting this finding, these genomes could represent genetically divergent ORFV strains that can infect both sheep and goats, or rather, that prefer only one of the two hosts despite remaining capable to also infect immunosuppressed individuals belonging to the other species. Such genome variants are more likely to occur and spread in areas where sheep and goats live in closer contact. In China, these two species are, in general, raised together, following an ancient practice that likely accounts for the comparable numbers of sheep and goats currently reported for this country [66].

The finding among the four genomes that can infect both sheep and goats of a strain isolated in Cuba would suggest that the raising of sheep and goats together is also a common practice in Cuba where this productive practice was described by Borroto et al. [67]. However, their historical connection dates to the arrival of Chinese workers in Cuba between 1847 and 1874 with a constant migration until 1959 [68,69]. In the future, the analysis of a higher number of ORFV genomes isolated for Cuba can shed further light on the presence of these rare type of genomes on the Caribbean island.

Noteworthily, in Sardinia, a Mediterranean island with a significantly lower goat population compared to sheep (about 3 million sheep and 280,000 goats in 2023) [70] and a few mixed farms, the study did not identify any genomes capable of infecting both sheep and goats. Despite representing 20% of the analyzed dataset, the ORFV strains isolated in Sardinia did not exhibit cross-species infectivity between sheep and goats. Based on the data obtained in the present study, it is not possible to rule out that the lack of this type of genomes in Sardinia may results from genetic drift that affected the early spread of ORFV on this island [29].

## 5. Conclusions

In the present study, the first deep phylodynamic inference with the highest level of resolution possible was performed for the ORFV genomes exclusive to sheep and goats. Results suggested that S and G genomes, which currently spread worldwide, differentiated in the second half of the 20th century producing several clusters likely endemic to specific geographic areas.

In this context, the presence in our research of the first Cuban genomes, which also represent the first ORFV genomic data from Central America, opens a new scenario for studying the genetic variation of this virus in isolated geographical areas in which phylodynamic patterns can be deeply inferred.

Interestingly, the outputs of the present studies indirectly contributed to shedding light on the origin of the ORFV lineages that can infect humans. Indeed, the “Sheep-to-Humans” genomes might have originated before sheep and goat genomes with which they share a common ancestor, presumably differentiated during the 19th century [21]. These lineages may have a reduced fitness and a weak potential for dispersal among hosts as suggested by their high genetic divergence and the lack of descendants during their evolutionary history [21]. For ”Sheep-to-Humans” genomes, infecting humans would represent the last stage of the virus life, which is likely followed by the extinction of the strain, as the possibility of transmission from human to human is extremely rare and has only been reported in a few cases associated with specific jobs or recreational exposures in children [71].

Furthermore, this study suggests, for the first time, the possible existence of viral strains capable of infecting both goats and sheep whose genomes deserve to be better investigated in the future. These quite uncommon ORFV variants could differentiate in areas of the world where sheep and goats are farmed together and could have a reduced fitness that prevented the differentiation of a large number of descendants.

In the future, the analysis of a larger number of ORFV genomes from geographic areas that have not been investigated to date, such as northern Europe, will enhance our understanding of the evolutionary history of this virus and its phylodynamic patterns across different geographic scales.

## Figures and Tables

**Figure 1 viruses-16-00158-f001:**
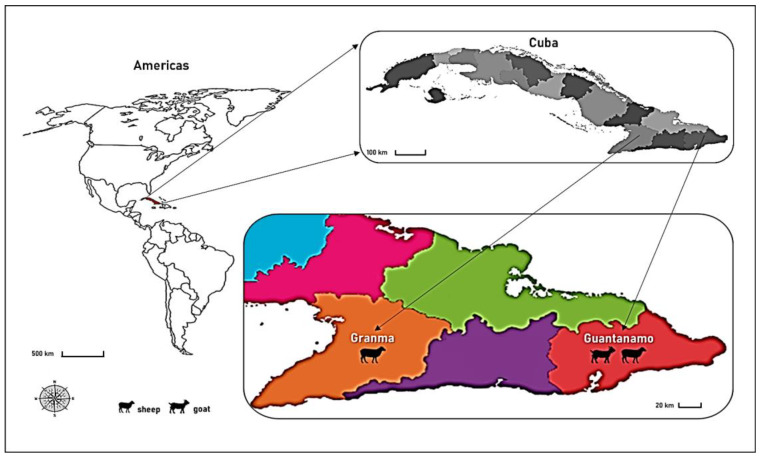
Sampling plan. Geographical distribution of the Cuban samples which were analyzed in the present study. Colors indicate the different provinces occurring in the geographic area.

**Figure 2 viruses-16-00158-f002:**
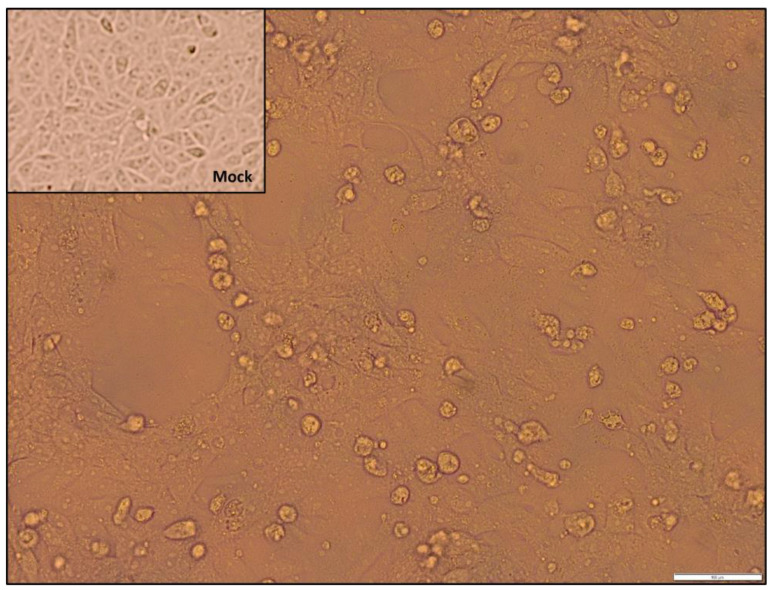
Viral replication on Vero E6 cell substrates. Comparison of cytopathic effect caused by specimen 33 at 72 h. Scale bar: 100 μM. Mock (non-infected cells).

**Figure 3 viruses-16-00158-f003:**
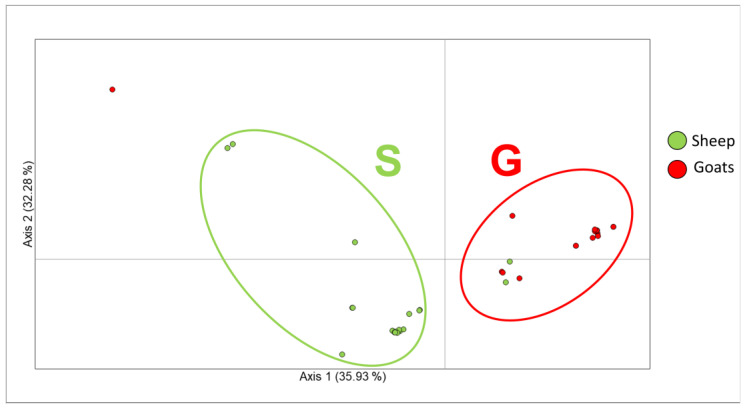
Principal Coordinate Analysis (PCoA) results. The plot evidences the genetic relationships among ORFV sheep (S) and goats (G) genomes based on a matrix of genetic distances. The genomes included within groups S and G are reported in Table 3. On the two axes, the percentage of genetic variation explained is indicated. The PCoA explains 68.21% of the total variability.

**Figure 4 viruses-16-00158-f004:**
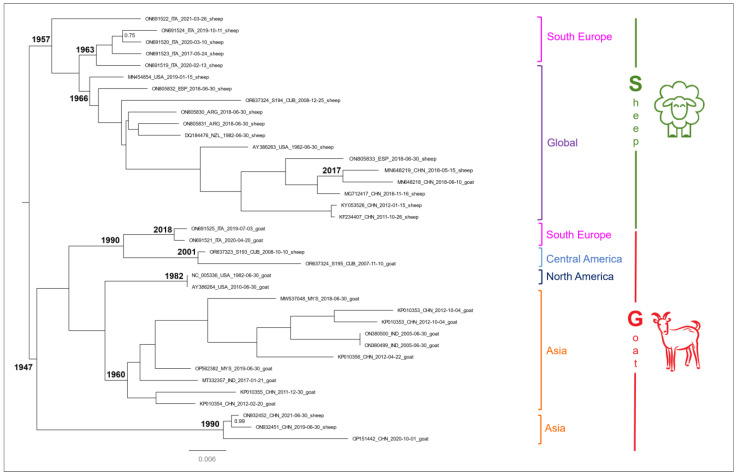
Bayesian phylogenetic tree based on ORF whole genomes. Node supports are expressed as posterior probabilities (PP). Only values of PP below 1 are reported in the tree. The median value of coalescence time is reported at the main nodes. Sequence labels include: GenBank accession number, host, international country code, and collection data.

**Figure 5 viruses-16-00158-f005:**
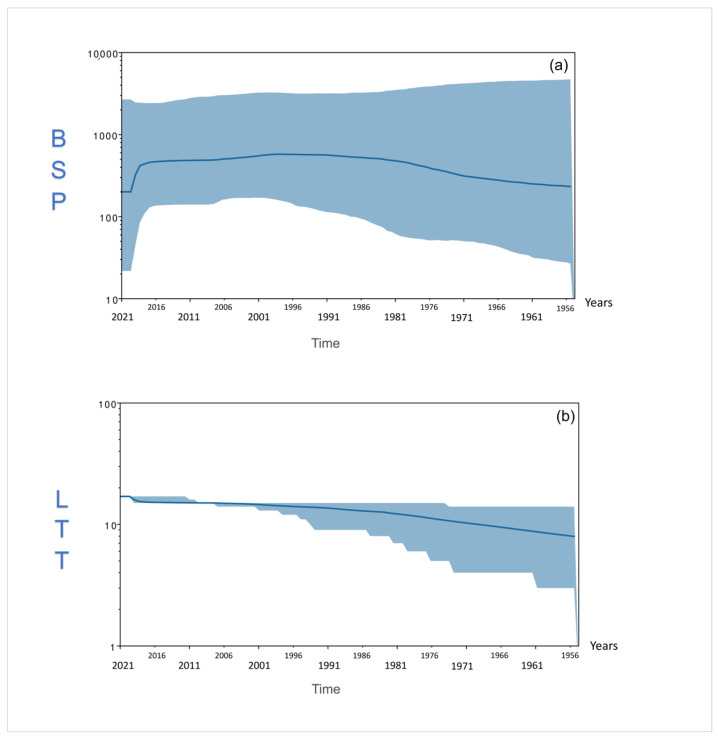
(**a**) Bayesian skyline plot (BSP) and (**b**) lineages through time (LTT) for ORFV goats’ (G) genomes. The effective population size and the number of lineages (*y*-axis) are shown as a function of time (*x*-axis). Values in the *y*-axis are expressed in a logarithmic scale and indicate the oscillation of genetic variability and number of lineages in the BSP and LTT graphs, respectively. Values in the *x*-axis are expressed in years. The thicker central line represents the median value of BSP and LTT, while the solid area represents the 95% high-posterior-density (HPD) region.

**Figure 6 viruses-16-00158-f006:**
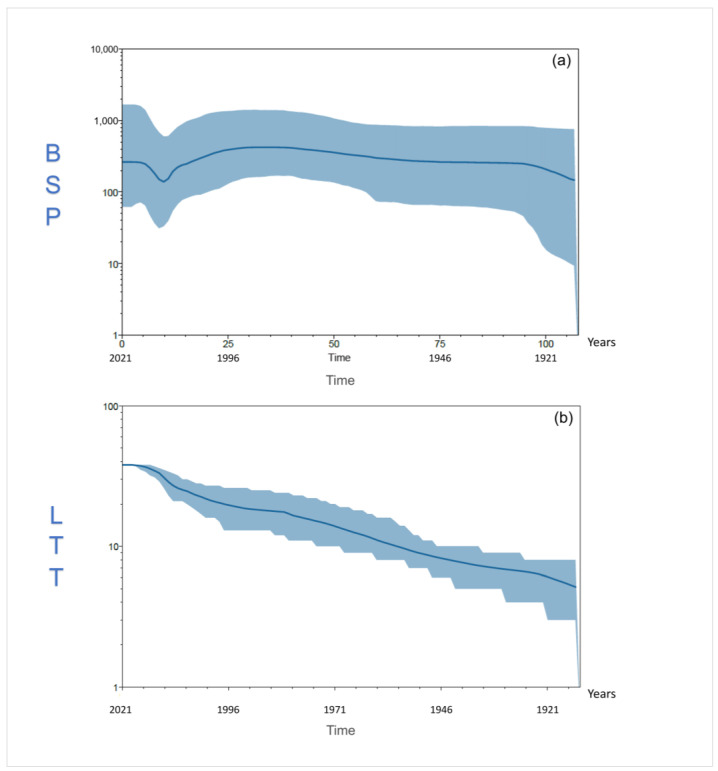
(**a**) Bayesian skyline plot (BSP) and (**b**) lineages through time (LTT) for ORFV sheep (S) genomes. The effective population size and the number of lineages (*y*-axis) are shown as a function of time (*x*-axis). Values in the *y*-axis are expressed in a logarithmic scale and indicate the oscillation of genetic variability and number of lineages in the BSP and LTT graphs, respectively. Values in the *x*-axis are expressed in years. The thicker central line represents the median value of BSP and LTT, while the solid area represents the 95% high-posterior-density (HPD) region.

**Table 1 viruses-16-00158-t001:** Sampling plan. The table reports data on the Cuban samples used in the present work. The collection site corresponds to the municipalities of the map in Figure 1.

Sample ID	Collection Site	Collection Date	Host
S44	Granma	25 December 2008	Sheep
S33	Guantanamo	10 November 2007	Goat
S45	Guantanamo	10 October 2008	Sheep

**Table 2 viruses-16-00158-t002:** Genomes metadata. ORF virus whole genomes from GenBank used in the present study.

Strain ID	GB #	Country	Collection Date	Host	Reference
NA1/11	KF234407	China	26 October 2011	Sheep	[17]
OVHN3/12	KY053526	China	15 January 2012	Sheep	Direct submission on GB
SY17	MG712417	China	16 November 2016	Sheep	Direct submission on GB
CL18	MN648219	China	15 May 2018	Sheep	Direct submission on GB
NP	KP010355	China	30 December 2011	Goat	[18]
GO	KP010354	China	20 February 2012	Goat	[18]
SJ1	KP010356	China	22 April 2012	Goat	[18]
YX	KP010353	China	4 October 2012	Goat	[18]
NA17	MG674916	China	2 December 2016	Goat	Direct submission on GB
GZ18	MN648218	China	10 June 2018	Goat	Direct submission on GB
MP	MT332357	India	21 January 2017	Goat	Direct submission on GB
Mukteswar_passage 9	ON380499	India	30 June 2005	Goat	Direct submission on GB
Mukteswar_ vaccine_passage 50	ON380500	India	30 June 2005	Goat	Direct submission on GB
HRE	ON805831	Argentina	30 June 2018	Sheep	Direct submission on GB
CHB	ON805830	Argentina	30 June 2018	Sheep	Direct submission on GB
NAV	ON805832	Spain	30 June 2018	Sheep	Direct submission on GB
ARA	ON805833	Spain	30 June 2018	Sheep	Direct submission on GB
NZ2	DQ184476	New Zealand	30 June 1982	Sheep	[42]
TVL	MN454854	USA	15 January 2019	Sheep	Direct submission on GB
OV-IA82	AY386263	USA	30 June 1982	Sheep	[43]
OV-SA00	NC005336	USA	30 June 1982	Goat	[43]
OV-SA00	AY386264	USA	30 June 2010	Goat	[20]
SC	ON932451	China	30 June 2019	Sheep	[19]
SC1	ON932452	China	30 June 2021	Sheep	[19]
nm-W	OP151442	China	1 October 2020	Goat	Direct submission on GB
UPM/HSN-20	MW537048	Malaysia	30 June 2018	Goat	[19]
UPM HSN-22	OP562382	Malaysia	30 June 2019	Goat	Direct submission on GB
S21	ON691523	Italy	24 May 2017	Sheep	[21]
S30	ON691525	Italy	3 July 2019	Goat	[21]
S27	ON691524	Italy	11 October 2019	Sheep	[21]
S6	ON691519	Italy	13 February 2020	Sheep	[21]
S10	ON691520	Italy	10 March 2020	Sheep	[21]
S15	ON691521	Italy	20 April 2020	Goat	[21]
S19	ON691522	Italy	26 March 2021	Sheep	[21]
S193	OR737323	Cuba	10 October 2008	Sheep	This study
S194	OR637324	Cuba	25 December 2008	Sheep	This study
S195	OR637325	Cuba	10 November 2007	Goat	This study

**Table 3 viruses-16-00158-t003:** Principal Coordinate Analysis (PCoA) results. The table shows the specific composition of the groups S and G that are reported in the Figure 3. Genomes labels include: GenBank accession number, host, international country code, and collection data.

**Cluster S**	
**Genome Label**	**Sampling Area**
ON932452_sheep_CHN_2021-06-30	China
ON932451_sheep_CHN_2019-06-30	China
KY053526_sheep_CHN_2012-01-15	China
KF234407_sheep_CHN_2011-10-26	China
ON805833_sheep_ESP_2018-06-30	Spain
MG712417_sheep_CHN_2016-11-16	China
MN454854_sheep_USA_2019-01-15	USA
ON691522_sheep_ITA_2021-03-26	Italy
OR637324_sheep_S194_44_CUB_2008-12-25	Cuba
ON805832_sheep_ESP_2018-06-30	Spain
ON805830_sheep_ARG_2018-06-30	Argentina
ON805831_sheep_ARG_2018-06-30	Argentina
DQ184476_sheep_NZL_1982-06-30	New Zealand
AY386263_sheep_USA_1982-06-30	USA
ON691520_sheep_ITA_2020-03-10	Italy
ON691524_sheep_ITA_2019-10-11	Italy
ON691519_sheep_ITA_2020-02-13	Italy
ON691523_sheep_ITA_2017-05-24	Italy
**Cluster G**	
**Genome Label**	**Sampling Area**
MN648219_sheep_CHN_2018-05-15	China
OR737323_Sheep_S193_45_CUB_2008-10-10	Cuba
ON691525_goat_ITA_2019-07-03	Italy
ON691521_goat_ITA_2020-04-20	Italy
MW537048_goat_MYS_2018-06-30	Malaysia
KP010353_goat_CHN_2012-10-04	China
KP010355_goat_CHN_2011-12-30	China
MT332357_goat_IND_2017-01-21	India
MG674916_goat_CHN_2016-12-02	China
AY386264_goat_USA_2010-06-30	USA
KP010356_goat_CHN_2012-04-22	China
KP010354_goat_CHN_2012-02-20	China
MN648218_goat_CHN_2018-06-10	China
ON380500_goat_IND_2005-06-30	India
OP562382_goat_MYS_2019-06-30	Malaysia
ON380499_goat_IND_2005-06-30	India
NC_005336_goat_USA_1982-06-30	USA
OR637325_goat_S195_33_CUB_2007-11-10	Cuba
**Outlier in the Plot**	
**Genome Label**	**Sampling Area**
OP151442_goat_CHN_2020-10-01	China

## Data Availability

The sequences of the ORFV whole genomes obtained during the present study are openly available in the GenBank nucleotide sequence database under the accession numbers OR737323-5.

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
