# Peer review of "The Global Evolutionary History of Orf Virus in Sheep and Goats Revealed by Whole Genomes Data"

_viruses, 2024, doi:10.3390/v16010158_

Round 1
Reviewer 1 Report
Comments and Suggestions for Authors
The manuscript deals with interesting aspects of Orf virus (ORFV) isolated from sheep and goats concerning virus evolution and host relationship derived from ORFV whole genome sequence (WGS) characteristics. The genomes of three recent ORFV isolates from Cuba representing Central America (Caribbean) are analysed in comparison with worldwide sheep and goat ORFV isolate WGS data from GenBank. Both, the WGS of Orf virus from Cuba and the comparative genome analysis are new and worth to be published.
However, there are several open questions, and some paragraphs contain statements/results that need to be explained and some of them afford correction.
1. The headline should be shortened and become more precise, e. g. “Orf virus in sheep and goats: phylodynamics revealed by analysis of whole genome sequence data including first isolate genomes from Central America”
2. Introduction and Material & Methods: the S for sheep origin and G for goat origin (line 86) must be used consequently. In Table 1 and Table 2 the goat specimen/isolate is named by suffix S but must be G. Concerning cell culture used for live ORFV isolation it is unclear why two non-host related cell lines are used? It is well known that ORFV can best be isolated and replicated in primary ruminant cells. Isolation and replication in cell lines and especially in alien cells are likely to cause ORFV genomic changes during an adaptation/ selection process. Viral DNA extraction and PCR: In line 141 it is said that DNA from 25 mg lesion tissue was extracted but the amount of DNA used in the PCR protocols and for sequencing is not indicated. It is not plausible why two different PCR protocols are used to target and amplify the VIR gene sequence. A robust PCR protocol should have been evaluated to secure only specific amplification of virus genome presence in samples and guarantee exclusion of virus genome negative samples. Table 2 Genomes metadata, continued page 6: The German strain D1701 is a non-suitable example for a European ORFV because it is highly attenuated and has been used as a vaccine, first published in 1981 (Mayr et al., J Med Vet B 28, 535-552). Unfortunately, WGS data of ORFV strains/Isolate genomes from Northern Europe are rare. However, for this comparative study the WGS data from isolate BO15 from 1996 (identical to BO29 accession No.# KF837136) as cited here in References (line 574) under Friederichs et al. (2013) will be best suited to replace strain D1701 in Table 2, Table 4 and Fig. 4.
3. Results: Figure 2 on page 7 should show cytopathic effect of goat ORFV specimen 33 in Vero and BEAS-2B cell lines. Virus-specific changes in cells can hardly be recognized, the picture of uninfected Vero cell control C resembles the infected cells A. For better visualisation a higher magnification is recommended otherwise the stated ballooning and syncytia are not recognizable. Usually, high degree of cytolysis and plaque formation will appear after 72 h in permissive cells and become more pronounced after a longer incubation period. In this context a convincing Figure 2 must be presented. On page 8 another PCR reaction targeting the B2L gene is mentioned (can be shifted to the Mat. & Meth. section) to monitor active virus replication in ?? cell substrates ?? (line 250) based on decreasing ct indicating increase of ORFV genome load. Under this aspect Table 3 is not meaningful and should be cancelled. On page 9 in Figure 3 the percentage numbers indicated on the x and y axis should be explained in the legend. In Figure 4 the paucity of ORFV WGS data from Northern Europe (only one from Germany to be corrected to strain BO15 from 1996, see 2.) reflects an overweight of Southern European WGS data (Italy, Spain) and thus is questioning the true positioning of European sheep/goat ORFV genomes. This fact should be expressed in the text describing the mid-point tree analysis (lines 298-352) and in the Discussion. In lines 332-333 as well as in line 343 and line 352 the reference to Table 3 is wrong, this table indicates B2L PCR data that should be cancelled as proposed before. To refer to Table 4 will be correct! In lines 348-352 the sub-cluster created for clade S (sheep ORFV genomes) including the German D1701 strain must be corrected in favour of isolate BO15 and therefore may show a new attribution since the date of isolation 1996 is exactly defined. Concerning the Bayesian skyline plots (Figures 5 and 6) the y-axis should be labelled e. g. by population and virus lineages, respectively. It will also be helpful to mark the year 2006 in the x-axis of Fig. 6a since it reflects the start of a decrease in the ORFV S genomes describes in text lines 365-370.
4. Discussion: the discussion is too long and must be shortened especially to avoid redundance and statements already published by the authors in 1921 and 1922. The evolutionary trend discussed starting in line 423 (page 15) covers an interesting feature in ORFV development over a long-time span. The influence of some worldwide viral epidemics (line 428) on goat and sheep population size should be concretized, e. g. the effect of Bluetongue Virus epidemics mentioned only by a citation notice (line 430). For the development of the goat G-genomes a similar discussion is focussed on a WHO report about goat mass mortality (lines 436-440) without precise information about the reasons and lacking a citation. The Discussion also includes the occurrence of so called “hybrid genomes” to represent genetically divergent ORFV (not strains) that can infect both sheep and goats (lines 458-462). It is plausible that such infections can occur when sheep and goats are raised together as practiced long time ago. However, it is expected that hybrids are molecularly precisely defined to communicate where in the genomes hybrid sequence spots can be found. It will also be important to know if certain genes are affected in hybrid genomes with consequences for the amino acid sequence. Citation of relevant literature even as example from other large DNA viruses will be informative.
5. Conclusions: in the conclusions the authors summarize their results and discussion well understandable. They also point out the zoonotic aspect of ORFV lineages and justify the weak potential of human ORFV isolates to undergo high genetic divergence. It is correctly stated that ORFV transmission among human patients is extremely rare as well as human-to-sheep/goat back transmission, concluding that zoonotic infection may be a dead end for ORFV.
Comments on the Quality of English LanguageSome sentences are very long and should be shortened.
Author Response
Reviewer 1
Dear Reviewer, please find below our response to your suggestions. Please consider that correction in the text made according to your advises were inserted using a RED font.
R1: The headline should be shortened and become more precise, e. g. “Orf virus in sheep and goats: phylodynamics revealed by analysis of whole genome sequence data including first isolate genomes from Central America”
A: We changed the title following the suggestions of the two Reviewers. For this reason, the Reviewer will find that we partially used the title that her/he proposed.
R1: the S for sheep origin and G for goat origin (line 86) must be used consequently. In Table 1 and Table 2 the goat specimen/isolate is named by suffix S but must be G.
A: In tables 1 and 2, in the first column the codes of sample/strain are reported without reference to the host for each sequence. For this reason, the capital letter “S”, before a number in the alphanumeric codes of our sequences, just stands for sample/strain. We would prefer to maintain the original code for our sequences to avoid problems with the identification of these sequences during our future studies. Please consider that in the analyses we used, for each sequence, a code that clearly refers to the host.
R1: Concerning cell culture used for live ORFV isolation it is unclear why two non-host related cell lines are used? It is well known that ORFV can best be isolated and replicated in primary ruminant cells. Isolation and replication in cell lines and especially in alien cells are likely to cause ORFV genomic changes during an adaptation/ selection process.
A: In general obtaining virus genome sequence directly from clinical samples is challenging due to the low load of virus genetic material compared to the host DNA. Propagation in cell culture is a step that is frequently included as an enrichment and cell lines were shown to provide faster viral replication prior to performing WGS. Vero cell lines are often used for virus isolation and culture and have been widely used for the study of many viruses including SARS-CoV-2. Our attempt to use cell lines instead of primary cultures of the host species, e.g. ovine/goat testis cells (cells that are, however, not the target of the virus), was aimed at verifying the susceptibility of a cell line such as VERO E6 widely included in WGS protocols of many viruses in order to achieve a standardization of the enrichment protocol. This allowed us to demonstrate an active viral replication in a highly permissive cell line, reducing the need to obtain cells from organ explants that are much more heterogeneous and complicated to manage in terms of biosafety and in vitro cultivation requirements (e.g keratinocytes). Orf virus is an epitheliotropic virus which is why it makes a lot of sense to use other epithelial cell types such as African green monkey kidney epithelial cells or Human bronchial epithelial cells for its propagation. The possibility that 3 passages in “alien” epithelial cell lines could cause changes on the genome of a double-stranded DNA virus, replicating in the cytoplasm and equipped with its own topoismerases, is certainly residual compared to the fact that passages in VEROE6 are widely used as pre-WGS enrichment for extreme error-prone RNA viruses such as coronaviruses. In addition, genomic modifications resulting from the propagation of parapoxviruses in cell culture mainly result in deletions of non-essential genes at the level of the terminal regions of the genome that were not found in sequencing.
R1: Viral DNA extraction and PCR: In line 141 it is said that DNA from 25 mg lesion tissue was extracted but the amount of DNA used in the PCR protocols and for sequencing is not indicated. It is not plausible why two different PCR protocols are used to target and amplify the VIR gene sequence. A robust PCR protocol should have been evaluated to secure only specific amplification of virus genome presence in samples and guarantee exclusion of virus genome negative samples.
A: In our previous study (Coradduzza et al 2021), we noticed that some samples with the first PCR protocol for the VIR gene did not amplify. This is probably due to genetic variability of viral strains; therefore, for that study, another PCR was designed that could amplify the samples that were not amplified by the first PCR. The use of both allows amplification of all samples increasing sensitivity.
R1: Table 2 Genomes metadata, continued page 6: The German strain D1701 is a non-suitable example for a European ORFV because it is highly attenuated and has been used as a vaccine, first published in 1981 (Mayr et al., J Med Vet B 28, 535-552). Unfortunately, WGS data of ORFV strains/Isolate genomes from Northern Europe are rare. However, for this comparative study the WGS data from isolate BO15 from 1996 (identical to BO29 accession No.# KF837136) as cited here in References (line 574) under Friederichs et al. (2013) will be best suited to replace strain D1701 in Table 2, Table 4 and Fig. 4.
A: We removed the German strain D1701 from the analyses since we agree with you on the potential issues associated with its use.
For what the isolate B015 concerns, its complete form/genome is not available. In both GenBank and the NCBI virus database, isolate B015 is only available for the genes ORF 011, also known as B2L (GB accession number: KF478796), and ORF 032 (GB accession number: KF478807), with the complete genome not being accessible. Conversely, the complete genome is available for the sample B029, but it corresponds to the human counterpart of the ovine isolate B015.
Since our research is focused on the reconstruction of the evolutionary dynamics of ovine and caprine ORFvirus strains, we decided to include in our dataset only these types of strains.
B015 and B029 are considered identical by Friederichs et al. since B015 has been mapped onto B029, which served as the reference genome. However, only the genes ORF11 and ORF32 have been investigated for these two strains making it possible to verify their identity solely on the bases of these two genome fragments. Consequently, we cannot take the risk of including B029 in the dataset as a substitute for B015 just hypothesizing their genomes are identical as it is possible that they differ for the other genes.
R1: Figure 2 on page 7 should show cytopathic effect of goat ORFV specimen 33 in Vero and BEAS-2B cell lines. Virus-specific changes in cells can hardly be recognized, the picture of uninfected Vero cell control C resembles the infected cells A. For better visualisation a higher magnification is recommended otherwise the stated ballooning and syncytia are not recognizable. Usually, high degree of cytolysis and plaque formation will appear after 72 h in permissive cells and become more pronounced after a longer incubation period. In this context a convincing Figure 2 must be presented.
A: We removed the part about Beas cells and replaced the photo. This choice was mandatory given that it has not been possible to repeat the infection on BEAS due to the timing of the review furthermore we choose to sequence only VERO replicated viruses. Now there is 1 photo related to the cytopathic effect on the Vero cells at 72h and a small image at the top left of photo where we show the uninfected cells.
R1: On page 8 another PCR reaction targeting the B2L gene is mentioned (can be shifted to the Mat. & Meth. Section) to monitor the active virus replication in cell substrates (line 250) based on decreasing CT indicating increase of ORFV genome load.
A: Done, we also added a reference.
R1: Under this aspect Table 3 is not meaningful and should be cancelled.
A: Done, Table 3 was deleted in the revised version.
R1: On page 9 in Figure 3 the percentage numbers indicated on the x and y axis should be explained in the legend.
A: Done.
R1: In Figure 4 the paucity of ORFV WGS data from Northern Europe (only one from Germany to be corrected to strain BO15 from 1996, see 2.) reflects an overweight of Southern European WGS data (Italy, Spain) and thus is questioning the true positioning of European sheep/goat ORFV genomes. This fact should be expressed in the text describing the mid-point tree analysis (lines 298-352) and in the Discussion.
A: Done. As above reported as answer to the comment 2, we removed D1701 since we agree with you on the potential issues associated with its use. For what concerns the overweight of sequences from southern Europe in the dataset, in the figure of the phylogenetic tree, we indicated the Italian sequences as “South Europe” in the revised version of the manuscript. Furthermore, we discussed this issue in the discussion.
R1: In lines 332-333 as well as in line 343 and line 352 the reference to Table 3 is wrong, this table indicates B2L PCR data that should be cancelled as proposed before.
A: The issue has been addressed and the table 3 removed.
R1: To refer to Table 4 will be correct!
A: Done.
R1: In lines 348-352 the sub-cluster created for clade S (sheep ORFV genomes) including the German D1701 strain must be corrected in favour of isolate BO15 and therefore may show a new attribution since the date of isolation 1996 is exactly defined.
A: Done. As above reported, the German D1701 has been removed from the latest version of the dataset. Unfortunately, as already mentioned in the response to comment 2, the complete genome of the strain B015 is not available neither in GenBank nor in the NCBI virus database and its identity with the strain B029 is confirmed only for 2 genes. For this reason, B015 cannot be included within our dataset.
R1: Concerning the Bayesian skyline plots (Figures 5 and 6) the y-axis should be labelled e. g. by population and virus lineages, respectively. It will also be helpful to mark the year 2006 in the x-axis of Fig. 6a since it reflects the start of a decrease in the ORFV S genomes describes in text lines 365-370.
A: Done. According to the Reviewer’s suggestion, we better labelled the axes of BSP and LTT. Furthermore, we improved the description of the analyses in the legend of the figures also following the request of another Reviewer. Indeed, in the revised version we better explained that values in y-axes are expressed in a logarithmic scale and indicate the oscillation of genetic variability and number of lineages in BSP and LTT, respectively. We indicated the years on x-axis to make easier to be interpretated the graph.
R1: the discussion is too long and must be shortened especially to avoid redundance and statements already published by the authors in 1921 (2021) and 1922 (2022). The evolutionary trend discussed starting in line 423 (page 15) covers an interesting feature in ORFV development over a long-time span. The influence of some worldwide viral epidemics (line 428) on goat and sheep population size should be concretized, e. g. the effect of Bluetongue Virus epidemics mentioned only by a citation notice (line 430). For the development of the goat G-genomes a similar discussion is focussed on a WHO report about goat mass mortality (lines 436-440) without precise information about the reasons and lacking a citation.
A: According to the suggestions of the Reviewer, we shortened the Discussion and included further references in order not to cite only our previous papers and also to provide references for our citation of viral epidemics. However, please consider that the description of the evolutionary history of ORFV is multifaceted and for this reason, the discussion of the manuscript requires detailed explanation of the outputs and reference to historical facts.
R1: The Discussion also includes the occurrence of so called “hybrid genomes” to represent genetically divergent ORFV (not strains) that can infect both sheep and goats (lines 458-462). It is plausible that such infections can occur when sheep and goats are raised together as practiced long time ago. However, it is expected that hybrids are molecularly precisely defined to communicate where in the genomes hybrid sequence spots can be found. It will also be important to know if certain genes are affected in hybrid genomes with consequences for the amino acid sequence. Citation of relevant literature even as example from other large DNA viruses will be informative.
A: In the revised version of the manuscript, we avoid the use of the term “hybrids” in order to avoid possible misinterpretation of our results description. We better explained that in this study we just report the occurrence of strain that can infect both sheep and goats without any inference to the molecular status of these strains that would deserve to be further investigated in the future.
R1: Some sentences are very long and should be shortened.
A: We revised the English form of the manuscript and shortened the sentences that were too long.
Reviewer 2 Report
Comments and Suggestions for Authors
viruses-2758272 Reviewer 1
As a continuation of their data published last year, in the presented work Coraduzza et al. pursued their investigation for the existence of phylogenetic S and G sub-clusters of Orf virus (ORFV), the type species of Parapoxvirus. The authors convincingly demonstrate the justification to phylogenetically subdivide ORFV into the 2 clades of sheep (S) and goat (G) derived virus isolates. Now the results of 38 known whole genome sequences (WGS) of ORFV were analyzed by modern phylogenetic programs with currently highest possible resolution. Moreover, for the first time a molecular dating of world-wide distributed ORFV was accomplished going back to the years 1947-1954.
Not to include human ORFV isolates for the WGS analysis was a clever and smart approach and this aspect is nicely discussed, comprehensible for the reader.
One might ask whether the term 'viral phylodynamics' is really correct for linking genomic data with evolutionary processes (host, year, livestock farming). Why not 'molecular dating'?
All in all this kind of genotyping and molecular dating nicely exemplifies a new perspective for the phylogenetic analysis, which is not only interesting for ORFV but also demonstrating a new, modern perspective to analyze phylogenetic relationship of viruses. Therefore, the presented ms is certainly worth to be published in this journal after major revisions.
Points of criticism
1. The title
1.1 'Walking Through the Evolutionary History of Orf Virus in Sheep and Goats: An Update on the Viral Phylodynamics with the First Whole Genomes from Central America' is unnecessarily long and thus, completely diffuses the subject of the work. Why not more concise as e.g.:
'The evolutionary history of worldwide ORFV in sheep and goats by molecular dating'. The new ORFV WGS from Cuba should be deleted, it is sufficient to be mentioned in the Abstract.
2. Abstract, needs major revision, the most important take-home message must be given clearly and concisely. Also revise the following confusing statements:
2.1 Line 27: worldwide diffusion, more accurate is widespread or worldwide distribution
2.2 Lines 29-32: It is not correct that only 'few fragmentary information' is available; in contrast quantities of reports describe genetic variations of ORFV. Additionally, the stated low genetic information is for sure not the reason to assess CE as a neglected disease?
2.3 Lines 31-35: Is the study really 'based' on the inclusion of 3 new ORFV isolates from Cuba; probably the authors mean 'included' and 'deeper' means 'enhance'. Also, why not saying the exact number of genomes instead of percentage?
3. Introduction
3.1 The introduction is generally well written. Some inaccuracies exist. E.g. line 76/77, I cannot agree that 'to the best knowledge' only a few WGS studies describe ORFV genomic differences, at least 20 papers are published on WGS of ORFV! Of course not all have to be cited, but that statement is wrong. Also at least 5 of the reports, not published by the authors, propose already the S and G clades, respectively.
3.2 Line 78: Instead of 'evidencing' it should be said e.g. 'indicating', in order to avoid misunderstanding of proving. This 'evidencing' or 'evidenced' is used many times throughout the text, which the authors should check more carefully for the meaning.
3.3 The part from line 93-106 should be shortened to the most important information needed in the Introduction.
3.4 Lines 90-91: This statement should be placed at the end of the Introduction.
3.5 Minor mistakes: Line 54 should be sealpox, line79 should be findings.
4. Material and Methods
4.1 Line 136: Without citation or more detailed explanation that description of 'in-house' B2L PCR is useless (see also comment below).
4.2 Lines 166-181: This part can be deleted, because this methods are already identically described in their former publication cited in (21).
4.3 Table 2 contains some sloppy errors: Why not all citations in parentheses, some 'goats' written in lowercases, 'Spagna' seems Italian for Spain.
4.4 The authors state the origin of the German strain D1701 as unknown. This is not true, and can be traced in publications of Mayr et al. and Rziha et al. In addition, to include that genome sequence is highly questionable for two reasons: (i) as published (Cottone et al. 1998, Rziha et al., 1999 to 2019) this virus represents a highly attenuated, multiply cell culture passaged, former anti-Orf vaccine virus containing several striking genomic rearrangements. Thus, what would be the value to include it in that kind of analysis?
(ii) As reported by Rziha et al. 2019, the submitted genome sequence of the used and by McGuire et al. not explained D1701 variant has several doubtful and unclear sequence ambiguities. Also questionable why incorporating it in such an otherwise excellent study.
4.5 Why not incorporating the German ORFV B015 described by Friederichs et al. 2015 and present in the GenBank?
5. Results
5.1 The chapter 3.1 describing the virus isolation is insufficient. Which samples or viruses have been isolated from the used cells? Notably, the applied cell lines Vero E6 as well as the BEAS-2B are very unusual or even not known for successful ORFV isolation. In addition, is it advantageous for the presented study to use viruses propagated in cells that are far from the natural hosts of ORFV?
The authors must explain their reasons and improve the quality of the presented result.
5.2 The Figure 2 is inadequate: Why not showing larger magnification, because at present differences between infected and not infected cells are poorly recognizable, if they exist! In Material and Methods the authors state the occurrence of an ORFV typical cpe, plaques and ballooning. However, cannot recognized in this figure, surprisingly small if any cpe after 72 hours of infection?
5.3 Next, the used 'in-house' PCR cannot be retraced, and Table 3 is useless and can be deleted. Moreover, the reason of applying the PCRs remains obscure. The need for additional VIR primers due to failure of PCR using VIR1 and VIR2, respectively? Why failure of this PCR, low sensitivity, not specific? Also results of this PCR are not shown? This obscurities must be dissolved.
5.4 Line 263: What is meant saying '38 131,449 bp long ORFV genome'? To my knowledge the analyzed 38 different WGS do not comprise identical bp numbers?
5.5 The PCoA analysis is convincing, but Table 4 should be revised: The outliers should be placed beneath the row 'Clusters S'. The two outliers are missing in Cluster G.
5.6 In all cases the 'German D1701' should be removed, lines 348-352 revised accordingly.
5.7 Concerning the BSP plots shown in Figures 5 and 6, respectively, the following critics or improving proposals: When describing the results, the results concluded from the curve shapes are needed to be explained shortly but understandable also for readers lesser informed in that field. For instance the meaning of the grey clouds remains as obscure as why in the G case (Fig. 5) the line is located in the cloud, but not in the S case (Fig. 6).
Next, what are the values plotted on the Y axis?
Also, the X axis should better indicate the annual figures (2000 etc.); plotting the years gone is very hard or even impossible to reproduce the presented results.
6. Discussion and Conclusion
In the main, the Discussion is well and in parts elegantly written. Particularly, it regards linking the genomic data with chronicle timeline, trade policy, and livestock farming in different countries of the world. Missing more countries from Northern Europe are of course limiting the conclusion.
6.1 However, unfortunately it is not a kind of novel, but a scientific paper. The main criticism refers to its length, and often too long sentences, which diffuse the arguments. Some conciseness would clearly improve it. Obviously, the authors preferably cite their own two publications up to 10-times or more. However, citations of other published reports regarding especially the S- and G cluster and sub-clusters are missing.
6.2 Lines 392-393: As also at other places, the authors should be more correct in writing ORFV-S or -G genomes; here is just one example of some more: 'genetic structuring between sheep and goat genomes'. But it must be 'structuring between genomes of ORFV isolated from sheep or goat'.
6.3 At some places the authors present or discuss statements without citations. For example, lines 426-435: Which viral epidemics are meant, what are the sources of the dates and numbers?
6.4 Hybrid genomes discussed from lines 458-489 and lines 527-532: What data presented allow that conclusion, except of the outliers in clades S and G, respectively? Hybrid genomes denote e.g. chimeric genomes assembled from different viruses or virus strains. What is the proof for that speculation that genomic parts or genes originate from an S and others from a G ORFV? Of course, it would be very interesting to investigate that hypothesis, e.g. with the presented outliers.
6.5 Moreover, what data presented unequivocally indicate or even prove that those 'hybrid viral strains' indeed infect sheep as well goats?
6.6 Exist studies of similar molecular dating for other viruses, too? If so, as experts in this field the authors should cite and briefly discuss.
6.7 Also the Conclusion part needs intense shortening to the main proven conclusions.
6.8 Would be very good to hear what limits of this study exist from the view of the authors? What's about ORFV WGS directly from the scab?

Minor editing of English language, as also indicated in the comments.
Author Response
REVISORE 2
Dear Reviewer, please find below our response to your suggestions. Please consider that correction in the text made according to your advises were inserted using a BLUE font.
R2: One might ask whether the term 'viral phylodynamics' is really correct for linking genomic data with evolutionary processes (host, year, livestock farming). Why not 'molecular dating'?
A: We reduced the use of the terms phylodynamics in the revised version of the manuscript to avoid possible issues derived from the description of our study. However, we would prefer to use this term since it exactly corresponds to the depicting of the evolutionary history of a virus with a specific focus on its temporal origin and the diffusion throughout space and time. The molecular dating represents one of the analytical tools used to reach the objective of phylodynamics. Actually, the term viral phylodynamics is the most comprehensive that can be used in this context.
R2: 1.1 'Walking Through the Evolutionary History of Orf Virus in Sheep and Goats: An Update on the Viral Phylodynamics with the First Whole Genomes from Central America' is unnecessarily long and thus, completely diffuses the subject of the work. Why not more concise as e.g.:
'The evolutionary history of worldwide ORFV in sheep and goats by molecular dating'. The new ORFV WGS from Cuba should be deleted, it is sufficient to be mentioned in the Abstract.
A: We changed the title following the suggestions of the two Reviewers. For this reason, the Reviewer will find that we partially used the title that her/he proposed.
R2: 2. Abstract, needs major revision, the most important take-home message must be given clearly and concisely. Also revise the following confusing statements:
A: Done.
R2: 2.1 Line 27: worldwide diffusion, more accurate is widespread or worldwide distribution
A: Done.
R2: 2.2 Lines 29-32: It is not correct that only 'few fragmentary information' is available; in contrast quantities of reports describe genetic variations of ORFV. Additionally, the stated low genetic information is for sure not the reason to assess CE as a neglected disease?
A: We modified this part accordingly.
R2: 2.3 Lines 31-35: Is the study really 'based' on the inclusion of 3 new ORFV isolates from Cuba; probably the authors mean 'included' and 'deeper' means 'enhance'. Also, why not saying the exact number of genomes instead of percentage?
A: Done.
R2: 3.1 The introduction is generally well written. Some inaccuracies exist. E.g. line 76/77, I cannot agree that 'to the best knowledge' only a few WGS studies describe ORFV genomic differences, at least 20 papers are published on WGS of ORFV! Of course not all have to be cited, but that statement is wrong. Also at least 5 of the reports, not published by the authors, propose already the S and G clades, respectively.
A: We changed this part, including in Introduction and Discussion a larger number of citations regarding molecular data on ORFV genomes and further citations on the previous reports on S and G clades. Please consider that since also another reviewer suggested to include the previous reports on S and G clades, these new citations could be included in the discussion in a RED font instead of BLUE.
R2: 3.2 Line 78: Instead of 'evidencing' it should be said e.g. 'indicating', in order to avoid misunderstanding of proving. This 'evidencing' or 'evidenced' is used many times throughout the text, which the authors should check more carefully for the meaning.
A: We reduced the use of “evidencing” in the revised version of the manuscript.
R2: 3.3 The part from line 93-106 should be shortened to the most important information needed in the Introduction.
A: Done.
R2: 3.4 Lines 90-91: This statement should be placed at the end of the Introduction.
A: We modified this whole part of the Introduction.
R2: 3.5 Minor mistakes: Line 54 should be sealpox, line79 should be findings.
A: done
R2: 4.1 Line 136: Without citation or more detailed explanation that description of 'in-house' B2L PCR is useless (see also comment below).
A: The proper citation was now added.
R2: 4.2 Lines 166-181: This part can be deleted, because this methods are already identically described in their former publication cited in (21).
A: Done. The indicated part has been strongly shortened.
R2: 4.3 Table 2 contains some sloppy errors: Why not all citations in parentheses, some 'goats' written in lowercases, 'Spagna' seems Italian for Spain.
A: This table has been revised and errors removed. The citation in parentheses correspond to paper which include the sequence, while the citation without parentheses correspond to the authors that directly deposited the sequence in Genbank without any paper corresponding.
R2: 4.4 The authors state the origin of the German strain D1701 as unknown. This is not true, and can be traced in publications of Mayr et al. and Rziha et al. In addition, to include that genome sequence is highly questionable for two reasons: (i) as published (Cottone et al. 1998, Rziha et al., 1999 to 2019) this virus represents a highly attenuated, multiply cell culture passaged, former anti-Orf vaccine virus containing several striking genomic rearrangements. Thus, what would be the value to include it in that kind of analysis?
(ii) As reported by Rziha et al. 2019, the submitted genome sequence of the used and by McGuire et al. not explained D1701 variant has several doubtful and unclear sequence ambiguities. Also questionable why incorporating it in such an otherwise excellent study.
A: We conducted a more in-depth investigation into the ambiguities of the D1701 genome, and we agree with the Reviewer regarding the necessity of its removal from the dataset. Consequently, in the revised version of the manuscript, analyses have been performed on a dataset that no longer includes D1701.
R2: 4.5 Why not incorporating the German ORFV B015 described by Friederichs et al. 2015 and present in the GenBank?
A: We agree with the reviewer regarding the importance of including the isolate B015 in the dataset; however, its complete genome is not available. Both GenBank and the NCBI virus database only provide sequences for the genes ORF 011 (also known as B2L with the GB accession number KF478796) and ORF 032 (GB accession number: KF478807) for isolate B015. But the complete genome of B015 is lacking.
Conversely, the complete genome is available for the sample B029, but it corresponds to the human counterpart of the ovine isolate B015 and we decided to include in our dataset only these types of strains.
Indeed, even if B015 and B029 are considered identical by Friederichs et al. since B015 has been mapped onto B029, which served as the reference genome. However, only the genes ORF11 and ORF32 have been investigated for these two strains making it possible to verify their identity solely on the bases of these two genome fragments.
R2: 5.1 The chapter 3.1 describing the virus isolation is insufficient. Which samples or viruses have been isolated from the used cells? Notably, the applied cell lines Vero E6 as well as the BEAS-2B are very unusual or even not known for successful ORFV isolation. In addition, is it advantageous for the presented study to use viruses propagated in cells that are far from the natural hosts of ORFV?
The authors must explain their reasons and improve the quality of the presented result.
A: In general obtaining virus genome sequence directly from clinical samples is challenging due to the low load of virus genetic material compared to the host DNA. Propagation in cell culture is a step that is frequently included as an enrichment and cell lines were shown to provide faster viral replication prior to performing WGS. Vero cell line are often used for virus isolation and culture and have been widely used for the study of many viruses including SARS-CoV-2. Our attempt to use cell lines instead of primary cultures of the host species e.g. ovine/goat testis cells (cells that are, however, not the target of the virus) was aimed at verifying the susceptibility of a cell line such as VERO E6 widely included in WGS protocols of many viruses in order to achieve a standardization of the enrichment protocol. This allowed us to demonstrate an active viral replication in a highly permissive cell line, reducing the need to obtain cells from organ explants that are much more heterogeneous and complicated to manage in terms of biosafety and in vitro cultivation requirements (e.g keratinocytes). Orf virus is an epitheliotropic virus which is why it makes a lot of sense to use other epithelial cell types such as African green monkey kidney epithelial cells or Human bronchial epithe-lial cells for its propagation. The possibility that 3 passages in “alien” epithelial cell lines could cause changes on the genome of a double-stranded DNA virus, replicating in the cytoplasm and equipped with its own topoismerases, is certainly residual compared to the fact that passages in VEROE6 are widely used as pre-WGS enrichment for extreme error-prone RNA viruses such as coronaviruses. In addition, the genomic modifications resulting from the propagation of parapoxviruses in cell culture mainly result in deletions of non-essential genes at the level of the terminal regions of the genome that were not found in sequencing.
R2: 5.2 The Figure 2 is inadequate: Why not showing larger magnification, because at present differences between infected and not infected cells are poorly recognizable, if they exist! In Material and Methods the authors state the occurrence of an ORFV typical cpe, plaques and ballooning. However, cannot recognized in this figure, surprisingly small if any cpe after 72 hours of infection?
A: We removed the part about Beas cells and replaced the figure. This choice was mandatory since it has not been possible to repeat the infection on BEAS due to the timing of the review. Furthermore, we choose to sequence only VERO replicated viruses. We now included in the revised version of the manuscript, a new figure related to the cytopathic effect on the Vero cells at 72h and a small image at the top left of figure where we show the uninfected cells.
R2: 5.3 Next, the used 'in-house' PCR cannot be retraced, and Table 3 is useless and can be deleted. Moreover, the reason of applying the PCRs remains obscure. The need for additional VIR primers due to failure of PCR using VIR1 and VIR2, respectively? Why failure of this PCR, low sensitivity, not specific? Also results of this PCR are not shown? This obscurities must be dissolved.
A: Citation was added and Table 3 deleted. In our previous study (Coradduzza et al 2021) we noticed that some samples with the first PCR protocol for the VIR gene did not amplify. This is probably due to genetic variability of viral strains; therefore, during that study, another PCR was designed that could amplify the samples that were not amplified by the first PCR. The use of both allows amplification of all samples.
R2: 5.4 Line 263: What is meant saying '38 131,449 bp long ORFV genome'? To my knowledge the analyzed 38 different WGS do not comprise identical bp numbers?
A: We better explained in the revised manuscript that we referred to the length of the sequence alignment.
R2: 5.5 The PCoA analysis is convincing, but Table 4 should be revised: The outliers should be placed beneath the row 'Clusters S'. The two outliers are missing in Cluster G.
A: We corrected the table 4 moving the row of the only outlier found by PCoA at the bottom of the table.
R2: 5.6 In all cases the 'German D1701' should be removed, lines 348-352 revised accordingly.
A: Done.
R2: 5.7 Concerning the BSP plots shown in Figures 5 and 6, respectively, the following critics or improving proposals: When describing the results, the results concluded from the curve shapes are needed to be explained shortly but understandable also for readers lesser informed in that field. For instance the meaning of the grey clouds remains as obscure as why in the G case (Fig. 5) the line is located in the cloud, but not in the S case (Fig. 6).
Next, what are the values plotted on the Y axis?
Also, the X axis should better indicate the annual figures (2000 etc.); plotting the years gone is very hard or even impossible to reproduce the presented results.
A: We better explained in the legend of the figure the meaning of the clouds and of the colored lines in the plots. Furthermore, according to the Reviewer’s suggestion, we better labelled the axes of BSP and LTT. In the revised version we better explained that values in y-axes are expressed in a logarithmic scale and indicate the oscillation of genetic variability and number of lineages in BSP and LTT, respectively. We indicated the years on x-axis to make easier to be interpretated the graph.
R2: 6.1 However, unfortunately it is not a kind of novel, but a scientific paper. The main criticism refers to its length, and often too long sentences, which diffuse the arguments. Some conciseness would clearly improve it. Obviously, the authors preferably cite their own two publications up to 10-times or more. However, citations of other published reports regarding especially the S- and G cluster and sub-clusters are missing.
A: According to the suggestions of the Reviewer, we shortened the Discussion and included further references in order not to cite only our previous papers and also to provide references for our citation of viral epidemics. However, please consider that the description of the evolutionary history of ORFV is multifaceted and for this reason, the discussion of the manuscript requires detailed explanation of the outputs and reference to historical facts.
R2: 6.2 Lines 392-393: As also at other places, the authors should be more correct in writing ORFV-S or -G genomes; here is just one example of some more: 'genetic structuring between sheep and goat genomes'. But it must be 'structuring between genomes of ORFV isolated from sheep or goat'.
A: We corrected these errors in the revised manuscript.
R2: 6.3 At some places the authors present or discuss statements without citations. For example, lines 426-435: Which viral epidemics are meant, what are the sources of the dates and numbers?
A: We added proper citations.
R2: 6.4 Hybrid genomes discussed from lines 458-489 and lines 527-532: What data presented allow that conclusion, except of the outliers in clades S and G, respectively? Hybrid genomes denote e.g. chimeric genomes assembled from different viruses or virus strains. What is the proof for that speculation that genomic parts or genes originate from an S and others from a G ORFV? Of course, it would be very interesting to investigate that hypothesis, e.g. with the presented outliers.
6.5 Moreover, what data presented unequivocally indicate or even prove that those 'hybrid viral strains' indeed infect sheep as well goats?
A: In the revised version of the manuscript, we avoid the use of the term “hybrids” in order to avoid possible misinterpretation of our results description. We better explained that in this study we just report the occurrence of strain that can infect both sheep and goats without any inference to the molecular status of these strains that would deserve to be further investigated in the future.
R2: 6.6 Exist studies of similar molecular dating for other viruses, too? If so, as experts in this field the authors should cite and briefly discuss.
A: We tried to modify the discussion to accomplish the request of the Reviewer. Indeed, we better explained that we used a combined approach that joins phylogenetic/phylodynamic and phylogeographic tools to reconstruct the evolutionary history of this virus. In this way, this kind of papers/studies would be proper of our group (and of course of the other groups with which we collaborate) and we would prefer to avoid further self-citations on ASFV, Monkeypox virus or Sars-Cov-2.
R2: 6.7 Also the Conclusion part needs intense shortening to the main proven conclusions.
A: Done. We shortened the Conclusion.
R2: 6.8 Would be very good to hear what limits of this study exist from the view of the authors? What's about ORFV WGS directly from the scab?
A: We better explained in the revised version of the manuscript that in the future the evolutionary trends we described in our study can be corroborated or changed by the analysis of a larger number of genomes with particular reference to strains from never investigated areas as North Europe.
For what concerns ORFV WGS directly from the scab, for diagnostic PCR we use DNA extracted from lesions but in this case, for whole genome sequencing we prefer to use viral supernatant to have a larger and cleaner amount of DNA. In general, obtaining virus genome sequence directly from clinical samples is challenging due to the low load of virus genetic material compared to the host DNA.
Round 2
Reviewer 1 Report
Comments and Suggestions for Authors
the revised manuscript is sufficiently improved and can be accepted for publication.
Reviewer 2 Report
Comments and Suggestions for Authors
After carefully reading the revised manuscript 'viruses-2758272-peer-review-v3' from Coradduzza et al., I can now fully agree with the revisions made, and thus, would like to recommend acceptance of it.